# Molecular analysis of secondary organic aerosol and brown carbon from the oxidation of indole

**Feng Jiang**[1,2], **Kyla Siemens**[3], **Claudia Linke**[1], **Yanxia Li**[1], **Yiwei Gong**[1], **Thomas Leisner**[1,5], **Alexander Laskin**[3,4], **and Harald Saathoff**[1]

[1]Institute of Meteorology and Climate Research, Karlsruhe Institute of Technology, 76344 Eggenstein–Leopoldshafen, Germany
[2]Institute of Applied Geosciences, Working Group for Environmental Mineralogy and Environmental System Analysis, Karlsruhe Institute of Technology, 76131 Karlsruhe, Germany
[3]Department of Chemistry, Purdue University, West Lafayette, Indiana 47907, United States
[4]Department of Earth, Atmospheric and Planetary Sciences, Purdue University, West Lafayette, Indiana 47907, United States
[5]Institute of Environmental Physics, Heidelberg University, 69120 Heidelberg, Germany

**Correspondence:** Feng Jiang (feng.jiang@kit.edu) and Harald Saathoff (harald.saathoff@kit.edu)

**Abstract.** Indole (ind) is a nitrogen-containing heterocyclic volatile organic compound commonly emitted from animal husbandry and from different plants like maize with global emissions of $0.1 \, \mathrm{Tg \, yr^{-1}}$. The chemical composition and optical properties of indole secondary organic aerosol (SOA) and brown carbon (BrC) are still not well understood. To address this, environmental chamber experiments were conducted to investigate the oxidation of indole at atmospherically relevant concentrations of selected oxidants (OH radicals and $O_3$) with or without $NO_2$. In the presence of $NO_2$, the SOA yields decreased by more than a factor of 2, but the mass absorption coefficient at 365 nm ($MAC_{365}$) of ind-SOA was $4.3 \pm 0.4 \, \mathrm{m^2 \, g^{-1}}$, which was 5 times higher than that in experiments without $NO_2$. In the presence of $NO_2$, $C_8H_6N_2O_2$ (identified as 3-nitroindole) contributed 76 % to all organic compounds detected by a chemical ionization mass spectrometer, contributing $\sim 50\%$ of the light absorption at 365 nm ($Abs_{365}$). In the absence of $NO_2$, the dominating chromophore was $C_8H_7O_3N$, contributing to 20 %–30 % of $Abs_{365}$. Indole contributes substantially to the formation of secondary BrC and its potential impact on the atmospheric radiative transfer is further enhanced in the presence of $NO_2$, as it significantly increases the specific light absorption of ind-SOA by facilitating the formation of 3-nitroindole. This work provides new insights into an important process of brown carbon formation by interaction of two pollutants, $NO_2$ and indole, mainly emitted by anthropogenic activities.

## 1 Introduction

Secondary organic aerosol (SOA) is generated by atmospheric oxidation of biogenic and anthropogenic volatile organic compounds (VOCs), and it has a profound impact on air quality, visibility, human health, and climate (Shrivastava et al., 2017; Hallquist et al., 2009). SOA is generally considered poorly absorbing and predominantly light-scattering material that leads to atmospheric cooling. However, there are coloured SOA compounds, known as brown carbon (BrC), which absorb solar radiation in the near-ultraviolet (UV) and visible spectral range contributing to warming effects (Moise et al., 2015; Laskin et al., 2015). BrC has a significant impact on climate forcing, accounting for 7 %–19 % of the light absorption by aerosols as estimated by global modelling study of Feng et al. (2013). In addition, Zeng et al. (2022) found that different geographic areas BrC can account for $\sim$ 7 %–48 % of the direct radiative forcing caused by both BrC and black carbon (BC).

Sources of BrC are mainly attributed to primary emission from biomass burning and fossil fuel combustion (Andreae and Gelencsér, 2006), complemented by secondary BrC compounds formed in the atmosphere as aged SOA of biogenic and anthropogenic origin (Xie et al., 2017; Laskin et al., 2014). Various studies have explored SOA and BrC from oxidation of aromatic VOCs, such as ethylbenzene (Yang et al., 2022), toluene (Lin et al., 2015; Y. X. Li et al., 2021), and naphthalene (Siemens et al., 2022; He et al., 2022). However, only a few investigations have examined SOA and BrC from oxidation of heterocyclic VOCs (Mayorga et al., 2022; Jiang et al., 2019; Montoya-Aguilera et al., 2017).

Indole, an important nitrogen-containing heterocyclic compound composed of fused benzene and pyrrole rings, is a significant VOC in the atmosphere. There are various natural and anthropogenic sources for atmospheric indole, including biomass burning (Laskin et al., 2009), animal husbandry (Yuan et al., 2017), agriculture plants like maize or rice (Erb et al., 2015; Skoczek et al., 2017), and tea manufacturing processes (Zeng et al., 2016). Global emissions and emission factors of indole are around $0.1\,\mathrm{Tg\,yr^{-1}}$ and $0.6\,\mathrm{\mu g\,m^{-2}\,h^{-1}}$, respectively (Misztal et al., 2015). This is already half of emissions of one of the most abundant amines, trimethylamine, with about $0.2\,\mathrm{Tg\,yr^{-1}}$ for global emissions (Yu and Luo, 2014). At selected locations, indole mixing ratios can reach around 1–2.7 ppb at daytime and 1.5–3.7 ppb at nighttime, which is around 10 times higher than isoprene levels during a spring flowering event at nighttime (Gentner et al., 2014). Despite its significant presence in the atmosphere, there are very limited studies investigating the formation of SOA and BrC from atmospheric oxidation of indole (referred to as ind-SOA hereafter).

An early study by Atkinson et al. (1995) investigated the gas-phase rate constants of indole with different oxidants. The study reported high rate constants for reactions of indole with hydroxyl (OH) and nitrate (NO$_3$) radicals $[(1.54 \pm 0.35) \times 10^{-10}$ and $(1.3 \pm 0.5) \times 10^{-10}\,\mathrm{cm^3\,molec.^{-1}\,s^{-1}}]$, respectively, while the reaction with ozone is much slower $(4.9 \pm 1.8) \times 10^{-17}\,\mathrm{cm^3\,molec.^{-1}\,s^{-1}}$ (Atkinson et al., 1995). A more recent study reported that the addition of OH radicals was the dominant pathway for indole reactions with N-(2-formylphenyl)formamide ($C_8H_7O_2N$) being an important product (Xue et al., 2022).

Environmental chamber experiments were conducted to study ind-SOA and its BrC components (Montoya-Aguilera et al., 2017; Baboomian et al., 2023). The SOA yield and mass absorption coefficient at 300 nm (MAC$_{300\,\mathrm{nm}}$) of ind-SOA were reported as $1.3 \pm 0.3$ and $\sim 2\,\mathrm{m^2\,g^{-1}}$, respectively. The absorption Ångström exponent (AAE) of ind-SOA were $6.8 \pm 0.2$ at 375–550 nm and $5.83 \pm 0.3$ at 315–450 nm, respectively (C. L. Li et al., 2021). The major monomer components of ind-SOA were identified as $C_8H_7O_3N$, isatin ($C_8H_5O_2N$), and isatoic anhydride ($C_8H_5O_3N$), while tryptanthrin ($C_{15}H_{10}O_2N_2$)

and indigo dye ($C_{16}H_{10}O_2N_2$) were the most abundant dimers (Montoya-Aguilera et al., 2017). Notably, nitroindole ($C_8H_6N_2O_2$) was one of the strongest chromophores in ind-SOA, produced in high abundance through both NO$_3$ radicals and OH radical oxidation in the presence of NO$_x$ (Baboomian et al., 2023). However, the chemical composition, formation mechanism, and optical properties of ind-SOA, including its BrC components formed at atmospherically relevant conditions, remain poorly understood.

In this study, we conducted simulation chamber experiments to investigate the chemical composition, formation mechanism, and optical properties of ind-SOA. The experiments were performed by ozonolysis and subsequent OH radical oxidation of indole at three simulated atmospheric conditions: a reference case of indole oxidation (REF), ind-SOA formed in presence of ammonium sulfate seed particles (AS), and ind-SOA formed in presence of both AS seed and NO$_2$ (AS-NO$_2$). We identified the major oxidized products of ind-SOA from the three experiments, measured its optical properties, and examine how NO$_2$ influences the atmospheric oxidation mechanisms leading to the formation of BrC chromophores.

## 2 Experimental methods

### 2.1 Particle generation and sampling

Experiments were conducted in the Aerosol Interaction and Dynamics in the Atmosphere (AIDA) simulation chamber operated at dark conditions, at relative humidity (RH) of $\sim 30\,\%$, and at room temperature ($\sim 303\,\mathrm{K}$), as shown in Table 1. The core measurement instrumentation used in this work is shown in Fig. S1 in the Supplement. The detailed description of the AIDA chamber, its operation modes, and the associated measurement techniques were published elsewhere (Gao et al., 2022; Saathoff et al., 2009). Briefly, ind-SOA was produced from ozonolysis of indole in the presence or absence of NO$_2$ and (NH$_4$)$_2$SO$_4$ seed particles, with experiment-specific conditions listed in Table 1. For the REF experiment (Fig. S2a), indole ($\geq 99\,\%$ purity, Sigma-Aldrich) was dissolved in in toluene ($> 99.9\,\%$ p.a. grade, Merk). The solution was used to generate an indole coating on a glass tube of 1.5 m length and 40 mm diameter by evaporating the toluene in the rotating tube. The indole was evaporated into the AIDA chamber with a flow of $0.01\,\mathrm{m^3\,min^{-1}}$ of synthetic air through the coated tube for 2–3 h obtaining indole mass concentrations of 90–150 $\mathrm{\mu g\,m^{-3}}$. The ozonolysis was started by adding 600–800 ppb of O$_3$. 2,3-dimethyl-2-butene (TME) was later injected and reacted with the excess of O$_3$ to form OH radicals (Lambe et al., 2007). For comparable levels of ozone and TME in the AIDA chamber, Salo et al. (2011) calculated OH radical levels of $0.2$–$1.0 \times 10^7\,\mathrm{molec.\,cm^{-3}}$ employing the Master Chemical Mechanism 3.1 (Bloss et al., 2005). Following injection of TME, rapid growth of ind-SOA was observed in

each of the experiments. To reduce the wall loss effect on ind-SOA formation, we inject ammonium sulfate (AS) seed particles. In these experiments (Fig. S2b and c), ammonium sulfate (Merck, 99.5 %) in ultra-pure water was aerosolized using an ultrasonic nebulizer (Synaptec), introduced into the AIDA chamber to reach mass loading of $\sim 50\,\mu g\,m^{-3}$ (Fig. S2b and c), before injecting indole. The number concentration and mobility mode size of the AS seed particles were $\sim 2000\,cm^{-3}$ and $\sim 230\,nm$, respectively. For $NO_2$-containing experiments referred as AS-$NO_2$ (Fig. S2c), $NO_2$ (99.5 % purity, Basi Schöberl GmbH) was injected into the chamber to reach $\sim 60\,ppb$ after finishing indole injection, followed by adding 100 ppb of $O_3$. However, indole was oxidized slowly in AS-$NO_2$ experiments, compared to REF and AS conditions. After 30 min, the products were further oxidized by adding more $O_3$ and TME. Please note that the difference of adding $O_3$ in AS-$NO_2$ experiments, compared with REF and AS experiments, was to investigate the reaction of indole with $NO_3$ radicals. However, this will not be discussed in this paper. The concentrations of $O_3$ and $NO_2$ in the chamber were recorded in real time by a gas monitor ($O_341M$ & AS32M, Environment S.A). Before starting experiments, the background samples from the AIDA chamber were collected on the filters. Compared to sample filters, the absorption of background filters from 240 to 800 nm only accounted for 1 %. Background measurements for both the gas and particle phase were performed before and after the first addition of indole to identify any contamination inside the chamber. The gas background levels were almost negligible for all experiments. For measurements by the chemical ionization mass spectrometer, most of the particle background signals were from filter matrix contaminations mainly due to fluorinated constituents. We subtracted the mass spectra of the background filter samples from those of the particle-loaded filter samples for the same experiments. This procedure has been described by Gao et al. (2022). After the mass concentrations of ind-SOA became stable inside the chamber, the aerosol samples were collected onto one Teflon filter (polytetrafluorethylene (PTFE), 1 μm, SKC Inc) and two parallel quartz filters (47 mm diameter, Whatman) for each experiment (Fig. S2).

## 2.2 Online measurement instrumentations

The particle number size distributions were monitored by a scanning mobility particle sizer (SMPS) containing a differential mobility analyser (DMA; 3071, TSI Inc.) connected to a condensation particle counter (CPC, 2772, TSI Inc.). In addition, particle number concentrations were measured by CPC (3022a, TSI Inc.). SMPS and CPC data analysis was also shown in a previous study (Gao et al., 2022). The chemical composition and aerodynamic size of ind-SOA were characterized by a high-resolution time-of-flight aerosol mass spectrometer (HR-ToF-AMS, Aerodyne Inc. hereafter AMS). Details about AMS calibration and data pro-

cessing are included in the Supplement (Sect. S1). The effective density ($\rho_{eff}$) was derived from the measurement of the vacuum aerodynamic diameter ($d_{va}$) obtained by AMS and the mobility equivalent diameter ($d_m$), see Figs. S11 and S12 (Kostenidou et al., 2007), as shown in the Supplement (Sect. S3). The indole concentration and lowly oxygenated gaseous oxidation products were measured by a proton-transfer reaction time-of-flight mass spectrometer (PTR-ToF-MS 4000, Ionicon Analytic GmbH), with specific details provided in the Supplement (Sect. S2). The SOA yields ($Y_{SOA}$) were calculated as $Y_{SOA} = \Delta SOA/\Delta VOC$, where $\Delta SOA$ values were inferred from the SMPS measurements and $\Delta VOC$ were measured by PTR-ToF-MS (see the Supplement, Sect. S4). The detailed information about the wall loss calculation is shown in Sect. S5 in the Supplement. During relatively short experimental time of < 200 min and due to the large size of the simulation chamber, particle losses contributed typically 4 % or less to the total SOA mass. In the presence of AS seed particles, the gas loss decreased by $\sim 6$ times. Since the wall losses of particles and trace gases were relatively low, we did not correct the gas and particle concentrations. Additionally, highly oxygenated gaseous oxidation products were measured with a filter inlet for gases and aerosols coupled to a high-resolution time-of-flight chemical ionization mass spectrometer (FIGAERO-HR-ToF-CIMS, Aerodyne Research Inc. hereafter CIMS) employing iodide ($I^-$) as the reagent ion.

## 2.3 Determination of optical properties of ind-SOA

Online analysis of the ind-SOA optical properties was performed using a photoacoustic spectrometer (PAS) operating at three wavelengths (405, 520, and 658 nm) (Linke et al., 2016). The mass absorption coefficient (MAC) was calculated as

$$MAC_{online}(\lambda) = \frac{\alpha(\lambda)}{C_{SOA}}, \tag{1}$$

where the $\alpha$ ($Mm^{-1}$) absorption is measured by PAS, and the $C_{SOA}$ ($\mu g\,m^{-3}$) is the SOA mass loading inferred from SMPS measurements.

Offline optical measurements were performed using Aqualog fluorometer (HORIBA Scientific, USA), which gives light absorption and excitation-emission spectra. Methanol soluble organic carbon (MSOC) was extracted from one quartz filter in each experiment with 5 mL of methanol (HPLC, Honeywell) using ultrasound sonication for 30 min. Please note that we used methanol and also acetonitrile (see next section) to extract the filter samples as to achieve comparability with previous work, i.e. by Montoya-Aguilera et al. (2017). Obtained extracts were filtered through a 0.45 μm polytetrafluoroethylene membrane into a glass vial to remove the insoluble material. The absorption was measured in the wavelength ranges of 239–800 nm with a 3 nm resolution

**Table 1.** Summary of indole SOA experimental conditions.

| Experiment ID | Indole (ppb) | NO$_2$ (ppb) | O$_3$ (ppb) | (NH$_4$)$_2$SO$_4$ seed | RH (%) | Temperature (K) | SOA density (g cm$^{-3}$) | SOA yield | SOA mass (µg m$^{-3}$) |
|---|---|---|---|---|---|---|---|---|---|
| REF | 20.5 | – | 698 | – | 29 | 303 | 0.8 ± 0.2 | 0.45 ± 0.1 | 45 ± 9.0 |
| AS | 24.5 | – | 700 | √ | 28 | 303 | 0.9 ± 0.2 | 0.44 ± 0.1 | 52 ± 10.4 |
| AS-NO$_2$ | 18.6 | 60 | 776 | √ | 29 | 303 | 1.3 ± 0.3 | 0.19 ± 0.04 | 21 ± 4.2 |

(Jiang et al., 2022). The MAC of the BrC fractions in the extracts were calculated as a function of wavelength according to Hecobian et al. (2010):

$$\text{MAC}_{\text{offline}}(\lambda) = (A_\lambda - A_{700}) \times \frac{V_{\text{extract}}}{V_{\text{air}} \times L \times C} \times \ln(10), \quad (2)$$

where $A_{700}$ and $A_\lambda$ are absorbance values measured by Aqualog, $V_{\text{extract}}$ (m$^3$) is the solvent volume, $V_{\text{air}}$ (m$^3$) is the sampling volume corresponding to the extracted filter, and $L$ is the optical path length of the quartz cuvette (1 cm); $C$ (µg m$^{-3}$) is the concentration of ind-SOA reported by SMPS, assuming that all absorbing ind-SOA components were dissolved in methanol.

## 2.4 Offline analysis by FIGAERO-CIMS and UPLC-PDA-HRMS

The Teflon filters analysed by FIGAERO-CIMS provide the organic molecular composition of the ind-SOA. The instrument and its modes of operation have been described in detail elsewhere (Jiang et al., 2022). Briefly, components of ind-SOA sample collected on the Teflon filter were desorbed by a flow of ultra-high-purity (99.9999 %) nitrogen gradually heated from 25 to 200 °C over the course of 35 min (Lopez-Hilfiker et al., 2014). The background filter sample was considered as a measurement background. The raw data were analysed using the toolkit Tofware (v3.1.2, Tofwerk, Thun, Switzerland, and Aerodyne, Billerica) with developed with the Igor Pro software (v7.08, Wavemetrics, Portland, OR). The molecular signals obtained were integrated as thermal desorption chromatograms after background subtraction. During the measurements, the mass resolution of FIGAERO-CIMS was relatively stable with about 4000 $m/\Delta m$. Since it was not possible for us to calibrate the sensitivities of all organic molecules, we assume the same sensitivity of 22 cps ppt$^{-1}$ for all compounds (Lopez-Hilfiker et al., 2016). Please note that the CIMS sensitivity of different organic compounds can vary by a few orders of magnitude. Part of these uncertainties were taken into account in the estimation of the overall uncertainties of CIMS concentrations (±60 %) following the approach by Thompson et al. (2017). Therefore, the mass fractions calculated from the CIMS measurements are actually percentages of the sum of the CIMS signals. Figure S10 shows the thermograms of four major compounds detected. For 3-nitroindole (C$_8$H$_6$O$_2$N$_2$), a single peak gives no indication for isomers or fragmentation.

However, broader thermograms, like those of C$_8$H$_7$O$_4$N, C$_8$H$_7$O$_3$N, and especially C$_8$H$_5$O$_3$N, may be caused by the presence of isomers of different volatility or thermal decomposition of larger molecules or oligomers (Lopez-Hilfiker et al., 2014). Furthermore, the thermograms are also influenced by the overall composition of the matrix on the filter, e.g. the ratio of the salts to the organics. This is shown in Fig. S13 with overall higher desorption temperatures for the experiment without ammonium sulfate. In this study we will focus on the molecular composition of ind-SOA but not discuss the details of its volatility.

A second quartz filter in each experiment was extracted with 8 mL acetonitrile (Optima LC/MS grade, Fisher Chemical) under sonication. The 8 mL of sample solution was then dried to 2 mL with pure nitrogen; 1 mL of the obtained concentrated solution was used to measure UV-Vis light absorption spectrum by Ocean Optics spectrophotometer. Another 1 mL of the concentrated solution was further dried to 200 µL and then analysed using an ultra-performance liquid chromatography (UPLC) system (Vanquish) coupled with a photodiode array detector (PDA), an electrospray ionization source (ESI) operated in negative mode, and a high-resolution mass spectrometer (HRMS) Q-Exactive HF-X hybrid quadrupole Orbitrap™ (all components from Thermo Scientific, Inc.). Detailed information of BrC characterization by UPLC-PDA-ESI/HRMS has been published elsewhere (Hettiyadura et al., 2021; Hettiyadura and Laskin, 2022). Briefly, aliquots of ∼ 0.3 µg of ind-SOA were injected into the UPLC for analysis. The ind-SOA components were separated on a reversed-phase polar C18 column (Luna Omega Polar C18 LC column; I. × I.D.: 50 × 2.1 mm; 1.6 µm particles; 100 Å pores; Phenomenex, Inc.) CEI using a 400 µL min$^{-1}$ binary solvent mixture containing water (A) and acetonitrile (B), with both solvents containing 0.1 % ($v/v$) formic acid. A 36 min LC gradient was programmed as follows: 0–0.5 min at 5 % of B, 0.5–26 min at a linear gradient to 100 % of B, 26–30.5 min B held at 100 %, 30.5–31 min decreased to 5 % of B, and 31–36 min held at 5 % of B to re-equilibrate the column. The UV-vis absorption spectra of eluted fractions were recorded using the PDA detector over the wavelength range of 300–680 nm. Mass spectra were acquired in negative mode over the $m/z$ range of 80–1200 Da at a mass resolution of $m/\Delta m = 240\,000$ at $m/z$ 200. The raw data were acquired using Xcalibur software (Thermo Scientific) and were further processed using MZmine 2.38 (He et

al., 2022). More information on the data analysis is shown in the Supplement.

## 3  Results and discussion

### 3.1  Yield and density of ind-SOA

Figure 1 presents the yields and densities of ind-SOA measured in three experiments of our study (REF, AS, and AS-NO$_2$). The REF and AS experiments exhibited the highest yields of ind-SOA at $0.45 \pm 0.1$ and $0.44 \pm 0.1$, respectively. The AS-NO$_2$ experiment showed the lowest yield of $0.19 \pm 0.04$ (Fig. S6 and Table 1). These observations are consistent with previously reported yields of SOA from oxidation of aromatic volatile organic compounds (AVOCs), which tend to show lower yields in the presence of NO$_x$ (NO$_2$ and NO) (Liu et al., 2021; Yang et al., 2022). The AVOCs oxidation products, organic peroxyl radicals (RO$_2$), mainly react with hydroperoxyl radicals (HO$_2$) under low-NO$_x$ conditions to form oxygenated low-volatility species which readily partition into the particle phase, providing growth of the SOA mass (Ng et al., 2007; Xu et al., 2014). However, in the presence of NO$_2$, gas-phase RO$_2$ intermediates can form peroxy nitrates (Orlando and Tyndall, 2012). In contrast, reactions of RO$_2$ with NO, RO$_2$, and NO$_3$ can produce alkoxy radicals which may undergo fragmentation. These volatile fragmented products remained in the gas phase, inhibiting the particle formation and growth. Notably, a previous study reported a higher yield of ind-SOA as $1.3 \pm 0.3$ under NO$_x$-free conditions with higher initial indole concentrations of 200 ppb (Montoya-Aguilera et al., 2017). In that study, formation of oligomer products of lower volatility was facilitated by higher VOC concentrations, resulting in greater partitioning into particle phase and higher yields. In our study, the highest density of ind-SOA was determined as $1.3 \pm 0.3$ g cm$^{-3}$ in the AS-NO$_2$ experiment, while the lowest density of $0.8 \pm 0.2$ g cm$^{-3}$ was observed in the REF experiment. These results suggest that products of large molecular weight were preferentially formed in ind-SOA under conditions of the AS-NO$_2$ experiment. This observation is consistent with the dominance of a 3-nitroindole (density: $1.4$ g cm$^{-3}$) as a significant component in ind-SOA products, accounting for 76 % as measured by FIGAERO-CIMS (Fig. 3c). Similarly, Ng et al. (2007) found that the effective density of m-xylene SOA formed in the presence of high-NO$_x$ and AS seed particles was $1.5$ g cm$^{-3}$, which was higher than the density of $1.3$ g cm$^{-3}$ observed under low-NO$_x$ conditions.

### 3.2  Optical properties of ind-SOA

As shown in Fig. 2, the indole precursor itself dissolved in methanol absorbs light only at wavelengths below 300 nm. However, all ind-SOA samples exhibit appreciable absorption values at longer wavelengths, as illustrated by high-est MAC$_{365\,\mathrm{nm,\ offline}}$ value of $4.3 \pm 0.4$ m$^2$ g$^{-1}$ measured in the AS-NO$_2$ experiment. Additionally, MAC$_{365\,\mathrm{nm,\ offline}}$ values in the AS and REF experiment were comparable but by a factor of 5 lower, $0.8 \pm 0.1$ and $0.7 \pm 0.1$ m$^2$ g$^{-1}$, respectively. Consistently, Montoya-Aguilera et al. (2017) also found that MAC of ind-SOA was $\sim 2$ m$^2$ g$^{-1}$ at 300 nm and $\sim 0.5$ m$^2$ g$^{-1}$ at 400 nm. Overall, MAC values reported here are higher than those published for secondary BrC generated by atmospheric oxidation of other VOCs, such as $0.29$ m$^2$ g$^{-1}$ (MAC$_{300-700\,\mathrm{nm}}$) for ethylbenzene-SOA formed at high-NO$_x$ conditions (Yang et al., 2022), $0.35$ m$^2$ g$^{-1}$ (MAC$_{375\,\mathrm{nm}}$) for pyrrole-SOA formed by NO$_3$ radical oxidation (Mayorga et al., 2022), and $\leq 0.01$ m$^2$ g$^{-1}$ (MAC$_{365\,\mathrm{nm}}$) for $\beta$-pinene-SOA (Yang et al., 2022). Therefore, ind-SOA is a comparably strong absorbing secondary BrC aerosol.

The highest MAC values were detected in the AS-NO$_2$ experiment, indicating that NO$_2$ had a significant effect on the formation of BrC chromophores and light-absorption by ind-SOA. Lin et al. (2015) also found that NO$_x$ affected the production of BrC chromophores in toluene-SOA, which were attributed to nitrophenols. Consistently, the online-measured MAC$_{405,\ \mathrm{online}}$ values by PAS in the AS-NO$_2$ experiment were also the highest values ($\sim 3.5 \pm 0.4$ m$^2$ g$^{-1}$), in comparison with MAC$_{405,\ \mathrm{online}}$ measured in the REF ($0.6 \pm 0.2$ m$^2$ g$^{-1}$) and AS ($0.9 \pm 0.3$ m$^2$ g$^{-1}$) experiments (Figs. 2b and S7). The MAC values in REF and AS were similar between online PAS and offline Aqualog measurements. However, for the AS-NO$_2$ experiment, PAS shows higher MAC values at 405 nm than the methanol extract. This difference may be caused by the uncertainties of both measurement techniques, but it may also be attributed to the difference between the absorption measurements of solution and the direct absorption measurement of particles in the air.

### 3.3  Chemical composition and chromophores of ind-SOA

Figure 3 presents the FIGAERO-CIMS mass spectra, O/C ratios, and fractions of C$_x$H$_y$O$_z$N$_{1-2}$ intensities for ind-SOA generated in the REF, AS, and AS-NO$_2$ experiments. In addition, the top 15 products are listed in Tables S1, S2, and S3 in the Supplement. In the REF and AS experiments (Fig. 3a and b), C$_8$H$_7$O$_4$N appears as the major product, accounting for $\sim 10$ % of all organic compounds detected by FIGAERO-CIMS. C$_8$H$_7$O$_3$N was the second most abundant product, accounting for 6 %–8 % of all organic compounds. C$_8$H$_5$O$_3$N (likely isatoic anhydride) was the third most abundant product, accounting for 5 %–6 % of all organic compounds detected by FIGAERO-CIMS. Interestingly, a new product, not appearing in the REF and AS experiments, was C$_8$H$_6$O$_2$N$_2$ (assigned as 3-nitroindole). Notably, in the AS-NO$_2$ experiment (Fig. 3c), 3-nitroindole significantly dominates ind-SOA products with 76 % of all organic compounds. Consistently, the 3-nitroindole exhibits the greatest cumulative intensity ($1 \times 10^9$ A.U.) measured

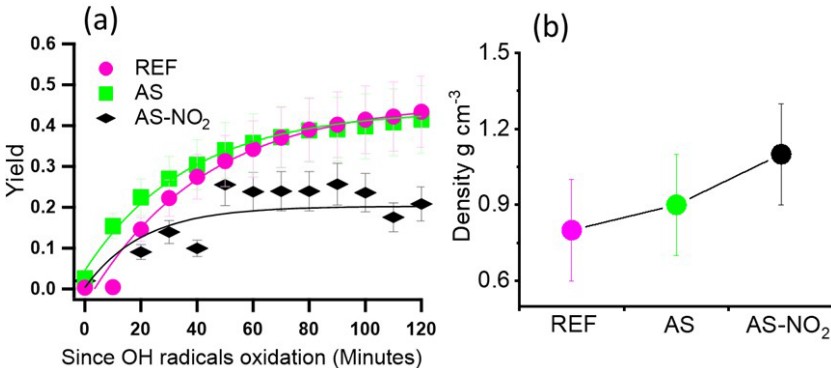

**Figure 1.** Evolution **(a)** of indole SOA yield with OH radical oxidation time and density **(b)** of indole SOA. The lines in **(a)** were fitted by exponential functions. REF (pink), AS (green), and AS-NO$_2$ (black). The yields were calculated under stable periods for 1 h.

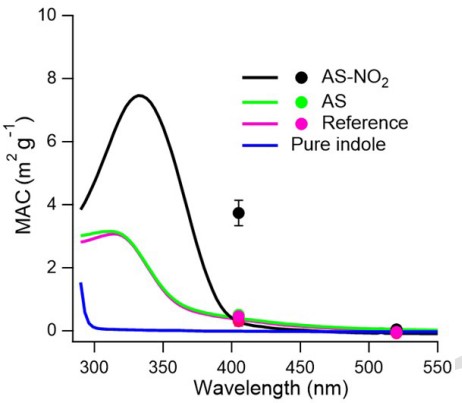

**Figure 2.** MAC of methanol-soluble ind-SOA between 290–550 nm from Aqualog measurements (lines). The MAC values of BrC aerosol particles at 405 and 520 nm measured by photoacoustic spectroscopy (circles). AS-NO$_2$ (black), AS (green), REF (pink), and pure indole (blue).

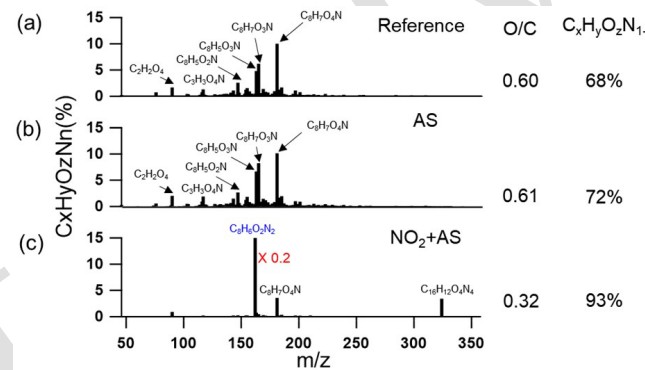

**Figure 3.** CIMS mass spectra of particle-phase ind-SOA (products generated in the REF, AS, and AS-NO$_2$ experiments). The CI source employs reactions of I$^-$ ions, which convert analyte molecules into [M+I]$^-$ ions. Legends above MS features correspond to neutral molecules. The blue legend indicates the most abundant features of C$_8$H$_6$O$_2$N$_2$. The mass fraction of C$_8$H$_6$O$_2$N$_2$ was multiplied by 0.2 since the fraction was too high. O/C ratios were calculated based on intensity-weighted sum of all compounds. The fraction of C$_x$H$_y$O$_z$N$_{1-2}$ of the total ion intensity is shown on the right.

by UPLC-PDA-MS (Fig. 4e). Furthermore, a dimer product, C$_{16}$H$_{12}$O$_4$N$_4$, accounted for 3.4 % of all organic compounds detected by FIGAERO-CIMS. Consistently, high intensities of large-weight fragment ions (C$_{13}$H$_6^+$, C$_5$H$_6$NO$_3^+$, and C$_7$H$_5^+$, etc.) measured by AMS could be fragments of 3-nitroindole or C$_{16}$H$_{12}$O$_4$N$_4$ (Fig. S8). Due to the high contribution of 3-nitroindole and C$_{16}$H$_{12}$O$_4$N$_4$, ind-SOA exhibits a lower O/C ratio (0.32) and a higher mass fraction (93 %) of nitrogen-containing molecules in the AS-NO$_2$ experiment relative to the REF and AS experiments (Fig. 3c). It is of note that the concentrations of VOC and oxidants in our experiments were more comparable with atmospheric conditions, which is unlike the previous studies discussed here (Montoya-Aguilera et al., 2017). Specifically, Montoya-Aguilera et al. (2017) reported that C$_8$H$_7$O$_3$N and C$_8$H$_5$O$_3$N were the first and second most abundant products in ind-SOA measured by LC-MS; 3-nitroindole was also identified but at a significantly lower abundance (Montoya-

Aguilera et al., 2017). The difference in relative abundance of these products in each study could be due to different experimental conditions and measurement techniques. In our study, lower fractions of products attributed to dimers and trimers were observed, while the previous study of Montoya-Aguilera et al. (2017) performed experiments at an order of magnitude higher indole concentration (200 ppb), resulting in higher observed fractions of dimers and trimers. A computational study of Xue et al. (2022) found that C$_8$H$_7$O$_2$N [N-(2-formylphenyl)formamide] was an important product from OH radical oxidation of indole. This molecule only accounted for 0.01 %–0.2 % of the organic mass in particle phase but had a higher fraction of 0.1 %–3 % in the gas phase measured by FIGAERO-CIMS.

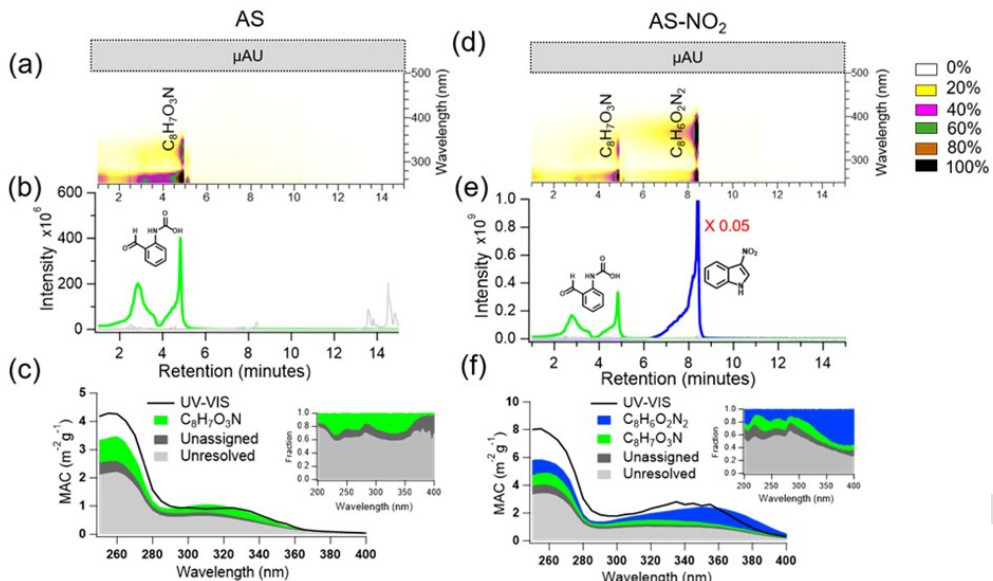

**Figure 4.** Molecular characteristics of individual components identified in the ind-SOA formed under AS and AS-NO$_2$ conditions. Panels **(a)** and **(d)** are the normalized UPLC-PDA chromatograms colour-mapped based on relative absorbance, with major chromophores labelled as neutral species. Panels **(b)** and **(e)** show a compilation of the selected extracted ion chromatograms (EICs) along with proposed molecular structures of the most abundant peaks. Panels **(c)** and **(f)** illustrate the MAC calculated from UV-visible spectrometer and PDA measurements. The PDA signal is grouped by contribution from unresolved chromophores (grey), unassigned chromophores (black), C$_8$H$_7$NO$_3$ (green), and C$_8$H$_6$N$_2$O$_2$ (blue).

The UPLC-PDA-ESI/HRMS results were used to identify individual BrC chromophores in the ind-SOA samples and to assess their contributions to the overall light-absorbing properties of ind-SOA formed in the three different experiments of our study. Please note that the MAC values determined from Aqualog and UPLC-PDA measurements show differences (Figs. 2 and 4). This could be caused by different solvent extraction, solution preparation, and instrumental differences. Figure 4 summarizes the BrC molecular characterization results observed for the AS and AS-NO$_2$ experiments, while those for the REF experiments are included in the Supplement (Fig. S9). In the ind-SOA sample from the AS experiment, the most abundant product, C$_8$H$_7$NO$_3$, shows a strong UV-vis feature absorbing in the range of 250–350 nm with $\sim 5 \times 10^4$ μA.U. signal intensity (Fig. 4a). This C$_8$H$_7$NO$_3$ product has also been detected as a high-intensity [M-H]$^-$ ion (Fig. 4b), consistent with the FIGAERO-CIMS measurements discussed above. Remarkably, this individual C$_8$H$_7$NO$_3$ chromophore accounts for 20 %–35 % of overall BrC absorption by the ind-SOA generated in the REF and AS experiments (Figs. 4c, S9c, Sect. S6), while its contribution was less prominent in the AS-NO$_2$ case. The BrC absorption in the AS-NO$_2$ case is heavily influenced by the 3-nitroindole chromophore, which exhibits a strong absorption band in the 300–400 nm range, resulting in a detected absorbance signal of $\sim 5.5 \times 10^4$ μA.U. (Fig. 4d). Here, the 3-nitroindole chromophore contributes even more significantly, accounting for $\sim 50$ % of the total BrC absorption (Fig. 4f). Consis-

tent with the FIGAERO-CIMS measurement, 3-nitroindole was detected as the very abundant [M-H]$^-$ ion, with a relative abundance approximately 2 orders of magnitude higher than all other components present in the mixture (Fig. 4e). It follows that NO$_2$ has a significant effect on the formation of 3-nitroindole, responsible for the BrC absorption in ind-SOA.

## 3.4 Reaction mechanism of the ind-SOA formation

The observed differences between ind-SOA products formed in different experiments provide insights into potential reaction mechanisms behind their formation, as summarized in Fig. 5. The reaction sequences start by abstraction of a hydrogen atom by either OH or NO$_3$ radicals, likely taking place in position 3 and leading to an indole radical (C$_8$H$_6$N$^\bullet$). Consistently, the attack of an electrophile at position 3 in indole generates a carbocation that did not disturb the aromaticity of the benzene ring (Sundberg, 1970). However, the attack of an electrophile at other positions in indole generates a carbocation that disrupted the aromatic character by delocalizing the positive charge over the benzene ring (Sundberg, 1970). Therefore, position 3 is a preferred site attacked potentially. In the absence of NO$_2$, C$_8$H$_6$O$_2$N radicals are formed by addition of O$_2$. Subsequently, the nitrogen-containing ring is opened by reaction with OH radicals, leading to the formation of an aldehyde (C$_8$H$_7$O$_2$N). In addition, indole can react with O$_3$ forming C$_8$H$_7$O$_3$N via ozone. Then the nitrogen-

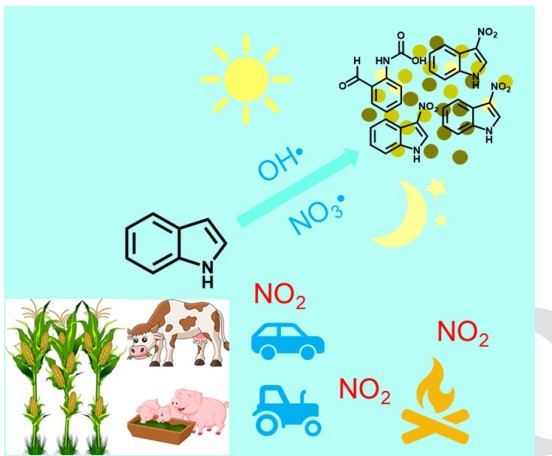

**Figure 5.** Proposed chemical reaction pathways leading to the major components of indole SOA observed in the particle phase including an efficient reaction with $NO_2$ to 3-nitroindole.

**Figure 6.** A schematic illustrating brown carbon formation by oxidation of indole emitted from maize and animal husbandry. TS1

containing ring is opened and also leads to formation of $C_8H_7O_2N$. Consistent with this, $C_8H_7O_2N$ was measured by CIMS, accounting for 0.2 % of all organic compounds detected by FIGAERO-CIMS. A computational study also predicted $C_8H_7O_2N$ as an important product from OH radical oxidation of indole (Xue et al., 2022). The aldehyde can then undergo further oxidation by OH radicals, forming acids like $C_8H_7O_3N$ and $C_8H_7O_4N$, as also measured by CIMS. This mechanism aligns with the formation of organic acids from the oxidation of aromatic VOC by OH radicals under low-$NO_x$ conditions, as reported in previous studies (Montoya-Aguilera et al., 2017; Siemens et al., 2022).

Remarkably, in the presence of $NO_2$, 3-nitroindole is formed in larger quantities and dominates ind-SOA products with 76 % of all organic compounds measured by CIMS. This is similar with OH radical induced oxidation of other aromatic compounds like toluene (Lin et al., 2015), naphthalene (Siemens et al., 2022), and pyrrole (Mayorga et al.,

2022), where nitro-aromatic compounds are formed in the presence of $NO_2$. The proposed mechanism is also consistent with the result that 3-nitroindole led to higher light absorption of ind-SOA in the presence of $NO_2$. Therefore, the presence of $NO_2$ during the formation of ind-SOA has a significant effect on the production of 3-nitroindole.

## 4   Atmospheric implications

This work provides valuable insights into the yield, chemical composition, formation mechanism, and optical properties of ind-SOA resulting from oxidation of indole in presence and absence of $NO_2$. In the absence of $NO_2$, the dominant products formed are $C_8H_7O_4N$ and $C_8H_7O_3N$. Furthermore, $C_8H_7O_3N$ appears as a major chromophore, contributing 20 %–30 % of the total light absorption of ind-SOA. However, in the presence of $NO_2$, a significant shift occurs, and 3-nitroindole becomes the dominant compound, comprising up to 76 % of the chemical composition measured by CIMS. Interestingly, the mass absorption coefficient at 365 nm ($MAC_{365}$) of ind-SOA in the presence of $NO_2$ is 5 times higher than that of ind-SOA formed under each of the other investigated conditions. Specifically, 3-nitroindole plays a critical role as the dominant chromophore in ind-SOA formed in the presence of $NO_2$, contributing to approximately 50 % of the total absorption at 365 nm. These results indicate that the presence of $NO_2$ during the formation of ind-SOA has a substantial impact on the light-absorbing properties of aerosol, primarily due to the facilitated formation of 3-nitroindole. Overall, this study complements and expands present understanding of ind-SOA formation mechanisms and underscores the significant influence of $NO_2$ on the chemical composition and the light-absorbing characteristics of ind-SOA.

Based on the yields and MAC determined in this study, we can estimate the potential absorption of ind-SOA formed in the presence of $NO_2$. Assuming atmospheric indole concen-

trations of approximately 1.8 during the daytime and 2.6 ppb during the nighttime, as reported for the spring flowering season (Gentner et al., 2014), we can calculate the potential absorption of atmospheric ind-SOA ($\mathrm{Abs_{ind\text{-}SOA}}$) as follows:

$$\mathrm{Abs_{ind\text{-}SOA}} = \mathrm{VOC} \times Y_{\mathrm{ind\text{-}SOA}} \times \mathrm{MAC}, \tag{3}$$

where VOC is the atmospheric indole concentration (Gentner et al., 2014), $Y_{\mathrm{SOA}}$ is the ind-SOA yield in the presence of $NO_2$, and MAC is the mass absorption coefficient of ind-SOA measured by PAS at 405 nm. Using this formula, we estimate the potential $\mathrm{Abs_{ind\text{-}SOA}}$ to be approximately $2.3 \pm 0.2\,\mathrm{Mm^{-1}}$ during the daytime and $3.2 \pm 0.3\,\mathrm{Mm^{-1}}$ during the nighttime. Notably, $\mathrm{Abs_{ind\text{-}SOA}}$ at both times exceeds the total absorption ($0.84 \pm 0.24\,\mathrm{Mm^{-1}}$) of ambient BrC measured during the summer at a rural background location in Melpitz, Germany (Moschos et al., 2018). Additionally, these calculated absorption values fall within the same range of total light absorption at 365 nm ($1.6 \pm 0.5\,\mathrm{Mm^{-1}}$ in summer and $2.8 \pm 1.9\,\mathrm{Mm^{-1}}$ in winter) observed for BrC aerosol in an urban area, the city of Karlsruhe (Jiang et al., 2022). These findings highlight that ind-SOA can have a significant impact on climate and air quality, particularly during the spring season or downwind of larger animal husbandries and maize fields. However, this may be limited to regions with corresponding emissions of both $NO_2$ and indole. The potential impact should be studied in atmospheric transport models using realistic emission scenarios.

**Data availability.** The data related to this article are accessible at KIT open data (https://doi.org/10.35097/1904, Jiang et al., 2024). Data are available upon request to the corresponding author. TS3

**Supplement.** The supplement related to this article is available online at: https://doi.org/10.5194/acp-24-1-2024-supplement.

**Author contributions.** FJ operated AMS and FIGAERO-CIMS; took the filter samples; analysed the filters by CIMS, UPLC-PDA-ESI/HRMS, UV-visible spectrometer, and Aqualog in the laboratory; performed the CIMS and UPLC-PDA-ESI/HRMS data analysis; produced all figures; and drafted the paper. KS operated UPLC-PDA-ESI/HRMS and UV-visible spectrometer and provided guidance for the associated tasks of data analysis and presentation. CL operated PAS and analysed the PAS data. YL operated PTR-ToF-MS and analysed the PTR-ToF-MS data. YG collected the filter samples. AM guided HRMS data analysis. TL gave general advice and comments for this paper. AL oversaw the UPLC-PDA-ESI/MS measurement and data interpretation. HS operated the AIDA simulation chamber and oversaw data analysis and interpretation. All authors provided suggestions for the data analysis, interpretation, and discussion and edited the manuscript.

**Competing interests.** At least one of the (co-)authors is a member of the editorial board of *Atmospheric Chemistry and Physics*.

The peer-review process was guided by an independent editor, and the authors also have no other competing interests to declare.

**Disclaimer.** Publisher's note: Copernicus Publications remains neutral with regard to jurisdictional claims made in the text, published maps, institutional affiliations, or any other geographical representation in this paper. While Copernicus Publications makes every effort to include appropriate place names, the final responsibility lies with the authors.

**Acknowledgements.** The authors gratefully thank the staff at IMK-AAF and KIT-IGG for providing substantial technical support. Feng Jiang thanks from China Scholarship Council (CSC) and KIT Grace for their support.

**Financial support.** Kyla Siemens and Alexander Laskin have received support from the U.S. National Science Foundation (award no. AGS-20239985). CE2

The article processing charges for this open-access publication were covered by the Karlsruhe Institute of Technology (KIT).

**Review statement.** This paper was edited by Theodora Nah and reviewed by three anonymous referees.

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

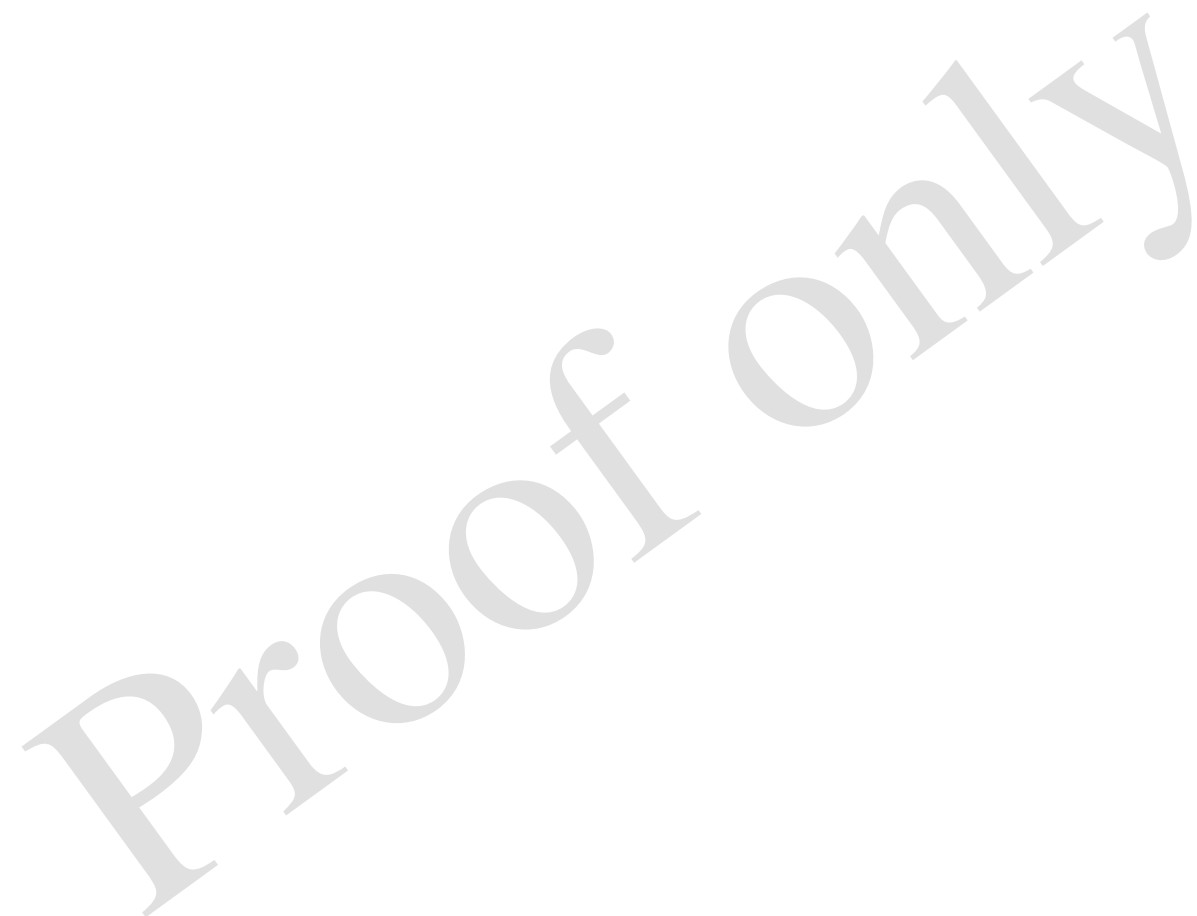

## Remarks from the language copy-editor

## Remarks from the typesetter

**TS1**    As this figure was also submitted as Fig. 6 in the preprint, we will have to ask the editor for approval. Please confirm that you meant that Fig. 6 should be deleted from the paper but should be used as a key figure for the article (key figures are shown in the right corner on the paper's website, e.g. https://acp.copernicus.org/articles/24/2267/2024/). Note that the caption will then also be deleted and not be shown, as key figures do not have a caption. Please provide a short explanation regarding this change that can be forwarded by us to the editor.