# Peer review of "Molecular Analysis of Secondary Organic Aerosol and Brown Carbon from the Oxidation of Indole"

_EGUsphere, 2023_

## Referee Comment (RC2)

**Comments to Jiang et al EGUsphere 2023:**

This manuscript by Jiang et al. explores chemical composition, formation mechanisms and optical properties of ind-SOA BrC produced from oxidation of indole in a environmental chamber at atmospherically relevent conditions with/without NO2. They observed that in the presence of NO2, the SOA yields decreased by more than a factor of two but the mass absorption coefficient of ind-SOA BrC at 365 nm was 5 times higher as compared to the ind-SOA BrC formed without NO2. The global emissions of Indole is half of one of the most abundant amines, i.e., trimethylamine. Despite its significant presence in the atmosphere, the chemical composition, formation mechanism, and optical properties of ind-SOA including its BrC remain poorly understood. The study is valuable for atmospheric chemistry and climate modelling community, particularly for areas with high Indole emissions, such as animal husbandry, maize and rice fields and tea manufacturing areas. This manuscript is well written, well-presented, and could be accepted for publication after considering the following comments:

Line 85-90: How did you make sure that there was no interaction between methanol/indole mixture? How will the volatilization of methanol will affect wall losses? Did you do blanks? Please elaborate.

Line 90: Did you use any tracer for OH concentration calculation? If yes, what tracer? Add a brief discussion about OH concentration calculation.

Section 3.2 (Line 195-210): You have used acetonitrile extracted ind-SOA BrC in UPLC-PDA analysis (section 3.3). However, BrC extraction efficiency in methanol and acetonitrile could be significantly different from each other. Why did you not compare ind-SOA BrC optical properties in methanol and acetonitrile?

Line 205-210: The MAC values in REF and AS were similar between online-PAS and offline-Aqualog measurements but not for AS-NO2. Why, elaborate?

Figure 4: How did you calculate the fraction of individual chromophores (known, unassigned, unresolved) to total indi-SOA absorption? Add a brief discussion.

Figure 4c: Typo-error "Unassiged"

Line 283: "However, in presence of NO2, a significant shift occurs, and 3-nitroindole becomes the dominant compound, comprising up to 76% of the chemical composition." I think it's 76% of the total CIMS species, not the total composition.

---

## Author Comment (AC1)

**Response to reviewers' comments on "Molecular Analysis of Secondary Organic Aerosol and Brown Carbon from the Oxidation of Indole" (egusphere-2023-1804)**

The authors kindly thank the reviews for the careful review of the manuscript, and the helpful comments and suggestions, which improve the manuscript a lot. All the comments are addressed below point by point, with our responses in blue, and the corresponding revisions to the manuscript in red. All updates of the original manuscript are marked in the revised version.

**Reviewer #1**

The manuscript by Jiang et al. investigates the oxidation of indole by selected oxidants (OH radicals and $O_3$) with/without $NO_2$. The authors report the chemical composition and optical properties of indole SOA (ind-SOA) under the investigated conditions. In the presence of $NO_2$, the ind-SOA yields decreased by more than a factor of two, but the mass absorption coefficient at 365 nm of ind-SOA was 5 times higher than that of the SOA form without $NO_2$. The global emission factors of indole could be around half of the emissions of the most abundant amines, trimethylamine. However, there are only limited studies investigating the formation of SOA and BrC from the oxidation of indole. Overall, this study would be a valuable addition to a better understanding of the ind-SOA formation mechanisms and the influence of $NO_2$ on the chemical composition and light-absorbing characteristics of ind-SOA. The results may be particularly important for areas with abundant indole emissions, such as large animal husbandries and maize fields. The manuscript is well-presented, and it could be accepted for publication after considering the comments below.

**General comments**

1. Line 90: To clarify how the OH concentrations were calculated, the authors could consider adding a few sentences explaining the methodology used.

We did not calculate the OH radical concentrations, but Salo et al. (2011) calculated the OH radical concentrations in the AIDA (Aerosol Interaction and Dynamics in the Atmosphere) by using the MCM 3.1 model (Bloss et al., 2005) for similar concentrations of $O_3$ and TME. We have explained this in the manuscript as follows:

"For comparable levels of ozone and TME in the AIDA chamber, Salo et al. (2011) calculated OH radical levels of $0.2–1.0\times10^7$ molecules $cm^{-3}$ employing the Master Chemical Mechanism 3.1 (Bloss et al., 2005)."

2. Lines 95-99 and Figure S2: The $O_3$ was injected into the chamber at around 600-800 ppb in the REF and seed experiments, while in the Seed-$NO_2$ experiment, it was initially added at around 100 ppb and then increased to 600-800 ppb after 30 minutes. The authors may want to provide an explanation for this difference.

In REF and seed experiments, the indole was oxidized by OH radical at high concentrations of ozone at around 600-800 ppb. In the seed-$NO_2$ experiment, $NO_2$ first reacted with $O_3$ and led to formation of the $NO_3$ radicals which reacted with indole. In a second step, the reaction products were oxidized by OH radicals leading to formation of secondary organic aerosol. The difference of the adding $O_3$ was to test the reaction ratio of indole with $O_3$ and $NO_3$ radicals.

"Please note that the difference of adding O$_3$ in AS-NO$_2$ experiments, compared with REF and AS experiments, was to investigate the reaction of indole with NO$_3$ radicals. However, this will not be discussed in this manuscript."

3. Line 100: It would be helpful if the authors could provide more information about the background samples and whether they would react with the reactants.

Light absorption from Aqualog measurements and chemical compositions from chemical ionization mass spectrometer (CIMS) measurements were corrected by subtracting background filter measurements. Compared to sample filters, the absorption from 240 nm to 800 nm only accounted for 1%. For CIMS measurement, the gas background was relatively low and particle background only contains fluorinated constituents, which don't interfere with sample analysis.

"Compared to sample filters, the absorption of background filters from 240 nm to 800 nm only accounted for 1%. Background measurements for both the gas and particle phase were performed before and after the first addition of indole to identify any contamination inside the chamber. The gas background levels were almost negligible for all experiments. For the measurements by the chemical ionization mass spectrometer, most of the particle background signals were from filter matrix contaminations mainly due to fluorinated constituents. We subtracted the mass spectra of the background filter samples from those of the particle-loaded filter samples for the same experiments. This procedure has been described by Gao et al., (2022)."

4. Line 117: Were the estimated trace gas and particle wall losses corrected?

We estimated the gas and particle wall loss by using model COSIMA. The wall loss of gas and particle mass was lower than 2%. Therefore, we did not correct the gas and particle concentrations.

"Since the wall losses of particles and trace gases were relatively low, we did not correct the gas and particle concentrations"

5. Lines 132 and 152: Why the methanol and acetonitrile were used to extract the filter samples for different analyses? It would be beneficial if the authors could explain their rationale for selecting these solvents and discuss any potential solvent effects.

Compared with the studies of the indole secondary organic aerosol by Montoya-Aguilera et al. (2017), the solution of methanol extraction was used to measure brown carbon aerosol. However, the solution of acetonitrile extraction was used to measure chromophores by using a high-performance liquid chromatography (HPLC) platform coupled to photo diode array (PDA) and high-resolution mass spectrometry (HRMS) detectors. Furthermore, methanol could induce solvent artifacts by reacting with conjugated carbonyl (Bateman et al., 2008; Chen et al. 2022). The acetonitrile, as an inert solvent, is an ideal for proper characterization of BrC chromophores (Chen et al. 2022).

"Please note that we used methanol and also acetonitrile (see next section) to extract the filter samples as to achieve comparability with previous work i.e. by Montoya-Aguilera et al. (2017)."

6. FIGAERO-CIMS part: The manuscript does not mention the mass resolution of the instrument used. Additionally, while the authors assumed a uniform sensitivity for different compounds, it is possible that sensitivities vary by order of magnitude. It would be helpful if

the authors could provide references from the literature supporting their assumption or consider rephrasing statements regarding "XXX% of CIMS detected compounds." Furthermore, it would be interesting to know if thermal desorption caused any fragmentation of the compounds and if multimodal thermograms were observed.

During the measurement, the mass resolution of FIGAERO-CIMS was relatively stable with about 4000 m/$\Delta$m. The most important issue for a quantitative mass detection efficiency for CIMS is its varying sensitivity for different chemical species. Sensitivity of Iodide-CIMS to individual compounds depends on their polarity and hydrogen bonding capability, and is strongly influenced by molecular geometry and steric factors (Caldwell, et al. 1989). Since it was not possible for us to calibrate the sensitivities of all organic molecules, we assume the same sensitivity of 22 cps/ppt for all compounds (cf. Lopez-Hilfiker et al. 2016). As shown in Figure S2 the major compounds ($C_8H_6O_2N_2$, $C_8H_7O_4N$, $C_8H_7O_3N$, and $C_8H_5O_3N$) detected by FIGAERO-CIMS exhibit only one peak in the thermogram and no substantial indication of fragmentation.

[Figure]

Figure S10. Thermograms of $C_8H_6O_2N_2$, $C_8H_7O_4N$, $C_8H_7O_3N$, and $C_8H_5O_3N$.

"During the measurements, the mass resolution of FIGAERO-CIMS was relatively stable with about 4000 m/$\Delta$m. Since it was not possible for us to calibrate the sensitivities of all organic molecules, we assume the same sensitivity of 22 cps/ppt for all compounds (Lopez-Hilfiker et al. 2016). As shown in Figure S2, the major compounds ($C_8H_6O_2N_2$, $C_8H_7O_4N$, $C_8H_7O_3N$, and $C_8H_5O_3N$) detected by FIGAERO-CIMS exhibit only one peak in the thermogram and no substantial fragmentation."

7. Line 172: What would be the reasons for the slightly lower SOA yield in the AS seed experiment than that in the REF experiment? Line 183: What is the seed concentration used in Montoya et al.? Would different seed concentrations play a role in the different yields?

The SOA mass was calculated by particle volume and density. Effective densities of indole SOA were derived from comparisons of the aerodynamic size distributions from AMS and the mobility size distributions from SMPS measurements (Saathoff et al. 2009; DeCarlo et al. 2004). The density calculation had relatively high uncertainties. The yield of indole SOA in the AS seed experiment was slightly lower than in the REF experiment but well within the combined uncertainties. Please note, that Montoya-Aguilera et al. (2017) didn't use seed aerosol. Generally, one would expect higher yields for experiments with seed aerosol as it reduces wall losses. However, wall losses were almost negligible in our study. Kamens et al. (2011) showed that experiments with higher initial seed concentrations (and particle phase water) generated more toluene-derived SOA than the lower seed experiments.

8. Figure 1b: When calculating the effective density of indole SOA by comparing the AMS and SMPS data, would the seed density affect the results? Was it excluded?

The density of indole SOA were derived from comparisons of the aerodynamic size distribution from AMS and the mobility size distribution from SMPS measurements. In the AS experiment, the major peaks of the size distributions were from pure indole SOA. Therefore, the seed particles do not affect the indole SOA density determination. However, in the AS-NO$_2$ experiment, the indole SOA density was calculated by the major peaks of coated particles. Therefore, we used an average particle density including the AS seed and the indole SOA coating. The particle mass concentration was calculated from particle volume and average density. The seed particle mass and volume can be measured by SMPS. We obtained the pure indole SOA mass and volume. Then we calculated the pure indole SOA density.

 "The effective density ($\rho$eff) was derived from the measurement of the vacuum aerodynamic diameter (dva) obtained by AMS and the mobility equivalent diameter (dm), see figures S11 and S12 (Kostenidou et al., 2007), as shown in the supplemental information (Sect. 3)."

[Figure]

Figure S11. Size distribution of indole SOA at AS experiment.

[Figure]

Figure S12. Size distribution of indole SOA at AS-NO₂ experiment.

9. Figure 3: It was mentioned in the figure caption that the Y-axis scale shows the fraction of $C_xH_yO_zN_{1-2}$ of the total ion intensity, but there are compounds without N atom shown in the Figure.

We agree that and modified the caption of Figure 3.

"The fraction of $C_xH_yO_zN_{1-2}$ of the total ion intensity is shown on the right."

10. Line 223: The author attributed the common ions $C_6H_4^+$ and $C_5H_3^+$ to be fragmented from 3-nitroindole or $C_{16}H_{12}O_4N_4$ (Figure S8), but these ions were also observed in REF and AS experiments.

We agree it and modified the sentence.

"Consistently, high intensities of large-weight fragment ions such as ($C_{13}H_6^+$, $C_5H_6NO_3^+$, and $C_7H_5^+$, etc.) measured by AMS could be fragments of 3-nitroindole or $C_{16}H_{12}O_4N_4$ (Fig. S8)."

11. Figure 4: Please check the caption about the description of the color used in the Figure. For example, "The unassigned chromophores (red)".

We agree and have corrected the caption in the Figure 4.

"…unassigned chromophores (black),…."

12. Line 249: 3-nitroindole contributed 76% of compound signals detected by a CIMS, and ~50% of the BrC absorption. Would this indicate there are compounds with low signal intensities that contribute even more than 3-nitroindole to the BrC absorption?

The mass fraction of 3-nitroindole was measured by a chemical ionization mass spectrometer. As discussed above, the relative abundance determined by this instrument has a substantial uncertainty as we only could assume an average sensitivity. The absorption fraction of 3-nitroindole was measured by HPLC-PDA-MS. The mass fraction values of 3-nitroindole were indeed different from absorption fraction values. This result could be caused by the two

different instruments. CIMS has varying sensitivity for different chemical species. Some chromophores can't be separated by HPLC and directly travel through the column. These could be the reasons why the mass fraction values are different with absorption fractions.

References

Bloss, C., Wagner, V., Jenkin, M. E., Volkamer, R., Bloss, W. J., Lee, J. D., Heard, D. E., Wirtz, K., Martin-Reviejo, M., Rea, G., Wenger, J. C., and Pilling, M. J.: Development of a detailed chemical mechanism (MCMv3.1) for the atmospheric oxidation of aromatic hydrocarbons, Atmos. Chem. Phys., 5, 641-664, 10.5194/acp-5-641-2005, 2005.

Bateman, A. P., Walser, M. L., Desyaterik, Y., Laskin, J., Laskin, A., and Nizkorodov, S. A.: The Effect of Solvent on the Analysis of Secondary Organic Aerosol Using Electrospray Ionization Mass Spectrometry, Environ. Sci. Technol., 42, 7341-7346, 10.1021/es801226w, 2008.

Caldwell, G., R. Renneboog, and P. Kebarle. 1989. 'GAS-PHASE ACIDITIES OF ALIPHATIC CARBOXYLIC-ACIDS, BASED ON MEASUREMENTS OF PROTON-TRANSFER EQUILIBRIA', Canadian Journal of Chemistry-Revue Canadienne De Chimie, 67: 611-18.

Chen, Kunpeng, Nilofar Raeofy, Michael Lum, Raphael Mayorga, Megan Woods, Roya Bahreini, Haofei Zhang, and Ying-Hsuan Lin. 2022. 'Solvent effects on chemical composition and optical properties of extracted secondary brown carbon constituents', Aerosol Science and Technology, 56: 917-30.

DeCarlo, P. F., J. G. Slowik, D. R. Worsnop, P. Davidovits, and J. L. Jimenez. 2004. 'Particle morphology and density characterization by combined mobility and aerodynamic diameter measurements. Part 1: Theory', Aerosol Science and Technology, 38: 1185-205.

Gao, L., Song, J., Mohr, C., Huang, W., Vallon, M., Jiang, F., Leisner, T., and Saathoff, H.: Kinetics, SOA yields, and chemical composition of secondary organic aerosol from β-caryophyllene ozonolysis with and without nitrogen oxides between 213 and 313 K, Atmos. Chem. Phys., 22, 6001-6020, 10.5194/acp-22-6001-2022, 2022.

Lopez-Hilfiker, F. D., S. Iyer, C. Mohr, B. H. Lee, E. L. D'Ambro, T. Kurten, and J. A. Thornton. 2016. 'Constraining the sensitivity of iodide adduct chemical ionization mass spectrometry to multifunctional organic molecules using the collision limit and thermodynamic stability of iodide ion adducts', Atmospheric Measurement Techniques, 9: 1505-12.

Montoya-Aguilera, J., J. R. Horne, M. L. Hinks, L. T. Fleming, V. Perraud, P. Lin, A. Laskin, J. Laskin, D. Dabdub, and S. A. Nizkorodov. 2017. 'Secondary organic aerosol from atmospheric photooxidation of indole', Atmospheric Chemistry and Physics, 17: 11605-21.

Kamens, R. M., Zhang, H., Chen, E. H., Zhou, Y., Parikh, H. M., Wilson, R. L., Galloway, K. E., and Rosen, E. P.: Secondary organic aerosol formation from toluene in an atmospheric hydrocarbon mixture: Water and particle seed effects, Atmospheric Environment, 45, 2324-2334, https://doi.org/10.1016/j.atmosenv.2010.11.007, 2011.

Kostenidou, E., Pathak, R. K., and Pandis, S. N.: An Algorithm for the Calculation of Secondary Organic Aerosol Density Combining AMS and SMPS Data, Aerosol Science and Technology, 41, 1002-1010, 10.1080/02786820701666270, 2007.

Saathoff, H., K. H. Naumann, O. Mohler, A. M. Jonsson, M. Hallquist, A. Kiendler-Scharr, T. F. Mentel, R. Tillmann, and U. Schurath. 2009. 'Temperature dependence of yields of secondary organic aerosols from the ozonolysis of alpha-pinene and limonene', Atmospheric Chemistry and Physics, 9: 1551-77.

Salo, K., M. Hallquist, A. M. Jonsson, H. Saathoff, K. H. Naumann, C. Spindler, R. Tillmann, H. Fuchs, B. Bohn, F. Rubach, T. F. Mentel, L. Muller, M. Reinnig, T. Hoffmann, and N. M. Donahue. 2011. 'Volatility of secondary organic aerosol during OH radical induced ageing', *Atmospheric Chemistry and Physics*, 11: 11055-67.

**Reviewer #2**

This manuscript by Jiang et al. explores chemical composition, formation mechanisms and optical properties of ind-SOA BrC produced from oxidation of indole in a environmental chamber at atmospherically relevent conditions with/without $NO_2$. They observed that in the presence of $NO_2$, the SOA yields decreased by more than a factor of two but the mass absorption coefficient of ind-SOA BrC at 365 nm was 5 times higher as compared to the ind-SOA BrC formed without $NO2$. The global emissions of Indole is half of one of the most abundant amines, i.e., trimethylamine. Despite its significant presence in the atmosphere, the chemical composition, formation mechanism, and optical properties of ind-SOA including its BrC remain poorly understood. The study is valuable for atmospheric chemistry and climate modelling community, particularly for areas with high Indole emissions, such as animal husbandry, maize and rice fields and tea manufacturing areas. This manuscript is well written, well-presented, and could be accepted for publication after considering the following comments:

**Major issues**

1. Line 85-90: How did you make sure that there was no interaction between methanol/indole mixture? How will the volatilization of methanol will affect wall losses? Did you do blanks? Please elaborate.

Sorry, there was a wrong description of the indole injection procedure. We have corrected this description. Actually, no methanol was involved. Instead, synthetic air was flushed through a glass tube coated with indole to inject it into the simulation chamber. The indole coating was generated by evaporating toluene from a solution of indole in toluene in the rotating glass tube. After adding indole into the chamber, we collected filter samples to measure the particle background. In addition, the gas phase background was measured by Iodide-CIMS and Proton Transfer Reaction – Mass Spectrometry (PTR-MS). During the measurement, we did not find any indication of toluene.

"For the REF experiment (Fig. S2a), indole ($\geq$ 99% purity, Sigma-Aldrich) was dissolved in toluene (>99.9% p.a. grade, Merk). The solution was used to generate an indole coating on a glass tube of 1.5 m length and 40 mm diameter by evaporating the toluene in the rotating tube. The indole was evaporated into the AIDA chamber with a flow of 0.01 $m^3$ $min^{-1}$ of synthetic air through the coated tube for 2-3 hours obtaining indole mass concentrations of 90–150 µg m$^{-3}$."

2. Line 90: Did you use any tracer for OH concentration calculation? If yes, what tracer? Add a brief discussion about OH concentration calculation.

We did not calculate the OH radical concentrations, but Salo et al. (2011) calculated the OH radical concentrations in the AIDA (Aerosol Interaction and Dynamics in the Atmosphere) by using the MCM 3.1 model at comparable concentrations of $O_3$ and TME. We have explained this in the manuscript as follows:

"For comparable levels of ozone and TME in the AIDA chamber, Salo et al. (2011) calculated OH radical levels of $0.2–1.0\times10^7$ molecules cm$^{-3}$ employing the Master Chemical Mechanism 3.1 (Bloss et al., 2005)."

3. Section 3.2 (Line 195-210): You have used acetonitrile extracted ind-SOA BrC in UPLC-PDA analysis (section 3.3). However, BrC extraction efficiency in methanol and acetonitrile

could be significantly different from each other. Why did you not compare ind-SOA BrC optical properties in methanol and acetonitrile?

We used the methanol extracts to allow direct comparison of our results with the study by Montoya-Aguilera et al. (2017). However, the acetonitrile extraction was used to measure chromophores by using a high-performance liquid chromatography (HPLC) platform coupled to photo diode array (PDA) and high-resolution mass spectrometry (HRMS), for which the filters are commonly extracted by using acetonitrile. We are aware that methanol could induce solvent artifacts by reacting with conjugated carbonyls (Bateman et al., 2008; Chen et al. 2022). In contrast, acetonitrile, as an inert solvent, is ideal for proper characterization of BrC chromophores (Chen et al. 2022). The data for the absorption spectra measured for both extracts are given in Figures 2 & 4.

4. Line 205-210: The MAC values in REF and AS were similar between online-PAS and offline-Aqualog measurements but not for AS-NO$_2$. Why, elaborate?

In the REF and AS experiment, the major chromophore was C$_8$H$_7$O$_3$N. However, in AS-NO$_2$ experiment, the major chromophore was C$_8$H$_6$O$_2$N$_2$ (3-nitroindole). Furthermore, 3-nitroindole becomes the dominant compound, comprising up to 76% of the chemical composition measured by CIMS in the AS-NO$_2$ experiment. For offline-Aqualog measurement, major absorption was measured from 3-nitroindole molecules dissolved in methanol. However, in the online-PAS measurement, 3-nitroindole was a component of the particle coating, which may cause differences between Aqualog and PAS measurements.

5. Figure 4: How did you calculate the fraction of individual chromophores (known, unassigned, unresolved) to total indi-SOA absorption? Add a brief discussion.

"We added relevant comments on HPLC-PDA data analysis in section 6 of the supplement."

"Fractions of MAC corresponding to each BrC feature (MAC$_{\lambda i}$) detected in the indole SOA (Figure 4c and f) are calculated using their relative absorptions and MAC$_\lambda$ as follows (Hettiyadura et al., 2021):

$$MAC_\lambda = MAC_\lambda \left( \frac{I_{\lambda_i} \times \Delta t_i}{I_{\lambda_i} \times \Delta t} \right)$$

where, I$_{\lambda i}$(μAU) is the averaged absorbance intensity of an individual BrC feature i and Δt$_i$ (min) is its time duration. I$_\lambda$ (μAU) is the averaged absorbance intensity across Δt =14 min of LC separation, which excludes the unresolved components eluted at (0 − 1 min). Unassigned fractions correspond to total absorption from 1-15 min, other than C$_8$H$_7$O$_3$N and C$_8$H$_6$O$_2$N$_2$ chromophores."

6. Figure 4c: Typo-error "Unassiged"

We have corrected the legend in the Figure 4c.

7. Line 283: "However, in presence of NO$_2$, a significant shift occurs, and 3-nitroindole becomes the dominant compound, comprising up to 76% of the chemical composition." I think it's 76% of the total CIMS species, not the total composition.

We agree and modify the sentence as follows:

"However, in presence of NO$_2$, a significant shift occurs, and 3-nitroindole becomes the dominant compound, comprising up to 76% of the chemical composition measured by CIMS."

Reference

Bloss, C., Wagner, V., Jenkin, M. E., Volkamer, R., Bloss, W. J., Lee, J. D., Heard, D. E., Wirtz, K., Martin-Reviejo, M., Rea, G., Wenger, J. C., and Pilling, M. J.: Development of a detailed chemical mechanism (MCMv3.1) for the atmospheric oxidation of aromatic hydrocarbons, Atmos. Chem. Phys., 5, 641-664, 10.5194/acp-5-641-2005, 2005.

Bateman, A. P., Walser, M. L., Desyaterik, Y., Laskin, J., Laskin, A., and Nizkorodov, S. A.: The Effect of Solvent on the Analysis of Secondary Organic Aerosol Using Electrospray Ionization Mass Spectrometry, Environ. Sci. Technol., 42, 7341-7346, 10.1021/es801226w, 2008

Chen, Kunpeng, Nilofar Raeofy, Michael Lum, Raphael Mayorga, Megan Woods, Roya Bahreini, Haofei Zhang, and Ying-Hsuan Lin. 2022. 'Solvent effects on chemical composition and optical properties of extracted secondary brown carbon constituents', Aerosol Science and Technology, 56: 917-30.

Hettiyadura, A. P. S., Garcia, V., Li, C., West, C. P., Tomlin, J., He, Q., . . . Laskin, A. (2021). Chemical Composition and Molecular-Specific Optical Properties of Atmospheric Brown Carbon Associated with Biomass Burning. Environmental Science & Technology. doi:10.1021/acs.est.0c05883

Montoya-Aguilera, J., J. R. Horne, M. L. Hinks, L. T. Fleming, V. Perraud, P. Lin, A. Laskin, J. Laskin, D. Dabdub, and S. A. Nizkorodov. 2017. 'Secondary organic aerosol from atmospheric photooxidation of indole', Atmospheric Chemistry and Physics, 17: 11605-21.

Salo, K., M. Hallquist, A. M. Jonsson, H. Saathoff, K. H. Naumann, C. Spindler, R. Tillmann, H. Fuchs, B. Bohn, F. Rubach, T. F. Mentel, L. Muller, M. Reinnig, T. Hoffmann, and N. M. Donahue. 2011. 'Volatility of secondary organic aerosol during OH radical induced ageing', Atmospheric Chemistry and Physics, 11: 11055-67.

---

## Referee Report (RR1)

This study by Jiang et al. investigated the formation, chemical composition, and optical properties of secondary organic aerosol (SOA) formed from ozonolysis and OH radical oxidation of iodole, one of the important nitrogen-containing heterocyclic VOCs. With the presence of NO2, iodole SOA formation potential decreased by a factor of two, but the light absorbing potential per mass was higher by a factor of 5. Using mass spectrometric techniques, the authors showed that the presence of NO2 shifted the iodole SOA formation chemical pathways, in which the formation of 3-nitroindole was significantly enhanced, hence affecting the overall SOA chemical composition and optical properties. This is an interesting study and valuable to the community of Atmospheric Chemistry and Physics. Overall, the authors appear to have addressed the reviewers' comments but there are some additional concerns associated with their responses.

Specific comments

- Figure S10: While the thermogram of C8H6O2N2 indeed looks a single peak, other compounds (i.e., C8H7O4N, C8H7O3N, and C8H5O3N) show much broader peaks, which are characteristics of multi-modal peaks. At least from calibrations, a single compound shows a very similar peak width (Lopez-Hilfiker et al., 2014; Stark et al., 2017), meaning that wider peaks indicate the presence of multiple peaks next to each other. This may be due to the presence of isomers with different volatilities or due to the effect of thermal decomposition. Therefore, I suggest rephrasing the sentence "… detected by FIGAERO-CIMS exhibit only one peak in the thermogram and no substantial fragmentation." 2-D thermogram (Wang and Ruiz, 2018; Takeuchi et al., 2022) may be a better way to illustrate the absence/presence of thermal decomposition in a holistic manner.
- Line 166ff: I agree with the authors that at this current moment, it is very difficult to do calibrations of all the detected organic compounds and it is not unreasonable to assume uniform sensitivity. However, I believe it is still important to note here that I-CIMS sensitivity could easily vary by a few order of magnitude reported in literature (Aljawhary et al., 2013; Lee et al., 2014), and therefore the reported fractions do not speak to the actual abundance. One idea is to explicitly say XXX% of the sum "signals" so it is obvious that percentage does not correspond to mass fraction.
- Line 200: I suggest rephasing this sentence. Reaction of gas-phase RO2 with NO2 produces peroxy nitrate (RO2NO2), which is thermally in equilibrium with RO2 and NO2 (Orlando and Tyndall, 2012). Production of fragments is typically from reaction of RO2 with NO, RO2, and NO3 that generate alkoxy radical (RO), which may undergo fragmentation pathways.
- Section 3.4 and Figure 5: Do you know how much indole reacted with ozone vs. OH vs. NO3 radical? From Figure S2, almost all indole appears to have reacted away before the addition of TME (source of OH radical), and I wonder if that means NO3 radical oxidation was the major oxidation pathway of iodole in AS-NO2 experiment, as opposed to ozonolysis.
- Table 1: I suggest adding [O3] as well because it is one important parameter of experimental condition. Also, the SOA mass concentration should be added next to the SOA yield because the SOA yield is a function of the SOA mass concentration and cannot be simply compared without it.
- SI Text 3 (SOA density): I suggest that the authors consider revising this paragraph incorporating their response to the comment 8 by the reviewer 1 to be complete.

References

Aljawhary, D., Lee, A. K. Y., and Abbatt, J. P. D.: High-resolution chemical ionization mass spectrometry (ToF-CIMS): application to study SOA composition and processing, Atmos Meas Tech, 6, 3211-3224, 10.5194/amt-6-3211-2013, 2013.

Lee, B. H., Lopez-Hilfiker, F. D., Mohr, C., Kurten, T., Worsnop, D. R., and Thornton, J. A.: An Iodide-Adduct High-Resolution Time-of-Flight Chemical-Ionization Mass Spectrometer: Application to Atmospheric Inorganic and Organic Compounds, Environ Sci Technol, 48, 6309-6317, 10.1021/es500362a, 2014.

Lopez-Hilfiker, F. D., Mohr, C., Ehn, M., Rubach, F., Kleist, E., Wildt, J., Mentel, T. F., Lutz, A., Hallquist, M., Worsnop, D., and Thornton, J. A.: A novel method for online analysis of gas and particle composition: description and evaluation of a Filter Inlet for Gases and AEROsols (FIGAERO), Atmos Meas Tech, 7, 983-1001, 10.5194/amt-7-983-2014, 2014.

Orlando, J. J. and Tyndall, G. S.: Laboratory studies of organic peroxy radical chemistry: an overview with emphasis on recent issues of atmospheric significance, Chem Soc Rev, 41, 6294-6317, 10.1039/c2cs35166h, 2012.

Stark, H., Yatavelli, R. L. N., Thompson, S. L., Kang, H., Krechmer, J. E., Kimmel, J. R., Palm, B. B., Hu, W. W., Hayes, P. L., Day, D. A., Campuzano-Jost, P., Canagaratna, M. R., Jayne, J. T., Worsnop, D. R., and Jimenez, J. L.: Impact of Thermal Decomposition on Thermal Desorption Instruments: Advantage of Thermogram Analysis for Quantifying Volatility Distributions of Organic Species, Environ Sci Technol, 51, 8491-8500, 10.1021/acs.est.7b00160, 2017.

Takeuchi, M., Berkemeier, T., Eris, G., and Ng, N. L.: Non-linear effects of secondary organic aerosol formation and properties in multi-precursor systems, Nat Commun, 13, 10.1038/s41467-022-35546-1, 2022.

Wang, D. S. and Ruiz, L. H.: Chlorine-initiated oxidation of n-alkanes under high-NOx conditions: insights into secondary organic aerosol composition and volatility using a FIGAERO-CIMS, Atmos Chem Phys, 18, 15535-15553, 10.5194/acp-18-15535-2018, 2018.

---

## Author Response (AR2)

**Response to reviewers' comments on "Molecular Analysis of Secondary Organic Aerosol and Brown Carbon from the Oxidation of Indole" (egusphere-2023-1804)**

The authors kindly thank the reviews for the careful review of the manuscript, and the helpful comments and suggestions, which improve the manuscript a lot. All the comments are addressed below point by point, with our responses in blue, and the corresponding revisions to the manuscript in red. All updates of the original manuscript are marked in the revised version.

**Editor**

Both the referees and I have gone through the authors' revisions and comments carefully. We are of the opinion that the FIGAERO thermograms need further analysis. Based on the FIGAERO thermograms that the authors' provided in the revised version, some of their thermograms could be fitted using multiple peaks (not single peaks). This indicates that they likely had thermal decomposition occurring for some of the thermograms of the products that they were tracking. This is especially obvious for the $C_8H_5O_3N$ thermogram, but the $C_8H_7O_4N$ and $C_8H_7O_3N$ thermograms could be fitted with multiple peaks as well. In addition, the peaks of these three thermograms were centered at high temperatures (>100 C), which is another indication of thermal decomposition. Based on the authors' replies to the comments in the initial round of reviews, it sounds like the authors may have prematurely discounted the possibility of thermal decomposition in their analysis. While this may not affect their overall conclusions, it is still important to get this analysis done correctly. The two referees have also made other minor comments that needs addressing before this manuscript can be accepted for publication.

Thank you for pointing to the analysis of the thermograms in our manuscript. We agree that the complexity of the thermograms needs to be addressed properly. Consequently, we have added more details on this in our answers to the reviewer comments below.

**Reviewer #1**

This study by Jiang et al. investigated the formation, chemical composition, and optical properties of secondary organic aerosol (SOA) formed from ozonolysis and OH radical oxidation of iodole, one of the important nitrogen-containing heterocyclic VOCs. With the presence of $NO_2$, iodole SOA formation potential decreased by a factor of two, but the light absorbing potential per mass was higher by a factor of 5. Using mass spectrometric techniques, the authors showed that the presence of $NO_2$ shifted the iodole SOA formation chemical pathways, in which the formation of 3-nitroindole was significantly enhanced, hence affecting the overall SOA chemical composition and optical properties. This is an interesting study and valuable to the community of Atmospheric Chemistry and Physics. Overall, the authors appear to have addressed the reviewers' comments but there are some additional concerns associated with their responses.

**Specific comments:**

1. Figure S10: While the thermogram of $C_8H_6O_2N_2$ indeed looks a single peak, other compounds (i.e., $C_8H_7O_4N$, $C_8H_7O_3N$, and $C_8H_5O_3N$) show much broader peaks, which are characteristics of multi-modal peaks. At least from calibrations, a single compound shows a very similar peak width (Lopez-Hilfiker et al., 2014; Stark et al., 2017), meaning that wider peaks indicate the presence of multiple peaks next to each other. This may be due to the

presence of isomers with different volatilities or due to the effect of thermal decomposition. Therefore, I suggest rephrasing the sentence "… detected by FIGAERO-CIMS exhibit only one peak in the thermogram and no substantial fragmentation." 2-D thermogram (Wang and Ruiz, 2018; Takeuchi et al., 2022) may be a better way to illustrate the absence/presence of thermal decomposition in a holistic manner.

Indeed, we didn't discuss the complexity of the thermograms sufficiently as we focused on the molecular composition of *ind*-SOA and potential reaction pathways. Broader thermograms, like those of $C_8H_7O_4N$, $C_8H_7O_3N$, and especially $C_8H_5O_3N$, may be caused by the presence of isomers of different volatility and thermal decomposition of larger molecules. Furthermore, the thermograms are also influenced by the overall composition of the matrix on the filter e.g. the ratio of the salts to the organics. To make the reader aware of this, we have modified the text in the manuscript as follows and we have also added 2-D thermograms as Figure S13. However, a detailed discussion of the SOA volatility is beyond the scope of our manuscript.

"Figure S10 shows the thermograms of four major compounds detected. For 3-nitroindole ($C_8H_6O_2N_2$) a single peak gives no indication for isomers or fragmentation. However, broader thermograms, like those of $C_8H_7O_4N$, $C_8H_7O_3N$, and especially $C_8H_5O_3N$, may be caused by the presence of isomers of different volatility or thermal decomposition of larger molecules or oligomers (Lopez-Hilfiker et al., 2014). Furthermore, the thermograms are also influenced by the overall composition of the matrix on the filter e.g. the ratio of the salts to the organics. This is shown in Figure S13 with overall higher desorption temperatures for the experiment without ammonium sulfate. In this study we will focus on the molecular composition of *ind*-SOA but not discuss the details of its volatility."

[Figure]

Figure S13. Two-dimensional thermograms of indole SOA at REF, AS, and AS-NO₂ experiments. The contour colors indicate normalized intensities.

2. Line 166: I agree with the authors that at this current moment, it is very difficult to do calibrations of all the detected organic compounds and it is not unreasonable to assume uniform sensitivity. However, I believe it is still important to note here that I-CIMS sensitivity could easily vary by a few order of magnitude reported in literature (Aljawhary et al., 2013; Lee et al., 2014), and therefore the reported fractions do not speak to the actual abundance. One idea is to explicitly say XXX% of the sum "signals" so it is obvious that percentage does not correspond to mass fraction.

We agree that it is important to point out the uncertainties of the CIMS measurements. Therefore, we added the following sentence to the manuscript.

"Since it was not possible for us to calibrate the sensitivities of all organic molecules, we assume the same sensitivity of 22 cps/ppt for all compounds (Lopez-Hilfiker et al., 2016). Please note that the CIMS sensitivity of different organic compounds can vary by a few orders of magnitude. Part of these uncertainties were taken into account in the estimation of the overall uncertainties of CIMS concentrations ($\pm$60%) following the approach by Thompson et al. (2017). Therefore, the mass fractions calculated from the CIMS measurements are actually percentages of the sum of the CIMS signals."

3. Line 200: I suggest rephasing this sentence. Reaction of gas-phase $RO_2$ with $NO_2$ produces peroxy nitrate ($RO_2NO_2$), which is thermally in equilibrium with $RO_2$ and $NO_2$ (Orlando and Tyndall, 2012). Production of fragments is typically from reaction of $RO_2$ with NO, $RO_2$, and $NO_3$ that generate alkoxy radical (RO), which may undergo fragmentation pathways.

We agree and modified the sentence as follows.

"However, in the presence of $NO_2$, gas-phase $RO_2$ intermediates can form peroxy nitrates (Orlando and Tyndall, 2012). In contrast, reactions of $RO_2$ with NO, $RO_2$, and $NO_3$ can produce alkoxy radicals which may undergo fragmentation."

4. Section 3.4 and Figure 5: Do you know how much indole reacted with ozone vs. OH vs. $NO_3$ radical? From Figure S2, almost all indole appears to have reacted away before the addition of TME (source of OH radical), and I wonder if that means $NO_3$ radical oxidation was the major oxidation pathway of iodole in AS-$NO_2$ experiment, as opposed to ozonolysis.

As shown in Figure S2, in the reference and seed particle experiments (S2a&b), the indole was depleted quickly by high concentrations of $O_3$. However, substantial SOA production only started after further oxidation of the reaction products by OH radicals. Atkinson et al. (1995) have reported a high-rate coefficient for reactions of indole with $NO_3$ radicals $(1.3 \pm 0.5) \times 10^{-10}$ cm$^3$ molecule$^{-1}$ s$^{-1}$], while the reaction with ozone is much slower $(4.9 \pm 1.8) \times 10^{-17}$ cm$^3$ molecule$^{-1}$ s$^{-1}$. Therefore, in AS-$NO_2$ experiment, with initially lower ozone levels (see Figure S2c), the indole was mainly oxidized by $NO_3$ radicals. Already a significant amount of SOA mass formed by $NO_3$ radical reactions. However, also in this case sub sequential reactions with OH radicals lead to a further substantial increase in SOA mass. The major product from oxidation of indole in presence of $NO_2$ was 3-nitroindole.

5. Table 1: I suggest adding [$O_3$] as well because it is one important parameter of experimental condition. Also, the SOA mass concentration should be added next to the SOA yield because the SOA yield is a function of the SOA mass concentration and cannot be simply compared without it.

We agree and modified Table 1 accordingly.

Table1. Summary of indole SOA experimental conditions.

| Experiment ID | Indole (ppb) | $NO_2$ (ppb) | $O_3$ (ppb) | $(NH_4)_2SO_4$ seed | RH (%) | Temperature (K) | SOA density (g cm$^{-3}$) | SOA yield | SOA mass (ug m$^{-3}$) |
|---|---|---|---|---|---|---|---|---|---|
| REF | 20.5 | _ | 698 | _ | 29 | 303 | 0.8 ± 0.2 | 0.45 ± 0.1 | 45 ± 9.0 |
| AS | 24.5 | _ | 700 | √ | 28 | 303 | 0.9 ± 0.2 | 0.44 ± 0.1 | 52 ± 10.4 |
| AS-$NO_2$ | 18.6 | 60 | 776 | √ | 29 | 303 | 1.3 ± 0.3 | 0.19 ± 0.04 | 21 ± 4.2 |

6. SI Text 3 (SOA density): I suggest that the authors consider revising this paragraph incorporating their response to the comment 8 by the reviewer 1 to be complete.

We agree and modified the Text 3 in the supplement. We added more information to describe how to calculate the density, especially for AS and AS-$NO_2$ experiments, as shown the below:

"As shown in Figure S11, in the AS experiment, the nucleation peaks of the size distributions were from pure indole SOA. Therefore, the seed particles do not affect the indole SOA density determination. However, in the AS-$NO_2$ experiment, the indole SOA density was calculated by the major peaks of coated particles as shown in Figure S12. Therefore, we used an average particle density including the AS seed and the indole SOA coating. The particle mass concentration was calculated from particle volume and average density. The seed particle mass and volume can be determined by SMPS. We obtained the pure indole SOA mass and volume. Then we calculated the pure indole SOA density."

Reference

Atkinson, R., Tuazon, E. C., Arey, J., and Aschmann, S. M.: Atmospheric and Indoor Chemistry of Gas-phase Indole, Quinoline, and Isoquinoline, Atmospheric Environment, 29, 3423-3432, 10.1016/1352-2310(95)00103-6, 1995.

Lopez-Hilfiker, F. D., Mohr, C., Ehn, M., Rubach, F., Kleist, E., Wildt, J., Mentel, T. F., Lutz, A., Hallquist, M., Worsnop, D., and Thornton, J. A.: A novel method for online analysis of gas and particle composition: description and evaluation of a Filter Inlet for Gases and AEROsols (FIGAERO), Atmospheric Measurement Techniques, 7, 983-1001, 10.5194/amt-7-983-2014, 2014.

Lopez-Hilfiker, F. D., Iyer, S., Mohr, C., Lee, B. H., D'Ambro, E. L., Kurten, T., and Thornton, J. A.: Constraining the sensitivity of iodide adduct chemical ionization mass spectrometry to multifunctional organic molecules using the collision limit and thermodynamic stability of iodide ion adducts, Atmospheric Measurement Techniques, 9, 1505-1512, 10.5194/amt-9-1505-2016, 2016.

Orlando, J. J., and Tyndall, G. S.: Laboratory studies of organic peroxy radical chemistry: an overview with emphasis on recent issues of atmospheric significance, Chemical Society reviews, 41 19, 6294-6317, 2012.

Thompson, S. L., Yatavelli, R. L. N., Stark, H., Kimmel, J. R., Krechmer, J. E., Day, D. A., Hu, W., Isaacman-VanWertz, G., Yee, L., Goldstein, A. H., Khan, M. A. H., Holzinger, R., Kreisberg, N., Lopez-Hilfiker, F. D., Mohr, C., Thornton, J. A., Jayne, J. T., Canagaratna, M., Worsnop, D. R., and Jimenez, J. L.: Field intercomparison of the gas/particle partitioning of oxygenated organics during the Southern Oxidant and Aerosol Study (SOAS) in 2013, Aerosol Science and Technology, 51, 30-56, 10.1080/02786826.2016.1254719, 2017.

**Reviewer #2**

The authors have generally responded well to the reviewer's comments. However, I would like to draw attention to a few areas that need further clarification or correction:

1. Figure S10: In the revised manuscript, the thermograms from Figaero CIMS measurements are presented. The thermogram for $C_8H_5O_3N$ displays multiple peaks. Despite this compound not being predominantly abundant in *ind*-SOA, and the possibility that the general findings remain valid, this particular detail merits closer scrutiny to ensure the robustness of the data.

Despite the complexity of the thermograms we still consider our interpretation of the molecular composition of the indole SOA as valid. It is beyond the scope of this manuscript to do a detailed discussion of the SOA volatility. However, to explain the differences in the thermograms we have added the following text and a 2-D representation of the thermograms.

"Figure S10 shows the thermograms of four major compounds detected. For 3-nitroindole ($C_8H_6O_2N_2$) a single peak gives no indication for isomers or fragmentation. However, broader thermograms, like those of $C_8H_7O_4N$, $C_8H_7O_3N$, and especially $C_8H_5O_3N$, may be caused by the presence of isomers of different volatility or thermal decomposition of larger molecules or oligomers (Lopez-Hilfiker et al., 2014). Furthermore, the thermograms are also influenced by the overall composition of the matrix on the filter e.g. the ratio of the salts to the organics. This is shown in Figure S13 with overall higher desorption temperatures for the experiment without ammonium sulfate. In this study we will focus on the molecular composition of *ind*-SOA but not discuss the details of its volatility."

[Figure]

Figure S13. Two-dimensional thermograms of indole SOA at REF, AS, and AS-NO₂ experiments. The contour colors indicate normalized intensities.

2. Line 95: "Following injection of OH radicals": It's stated that OH radicals were injected, which is inaccurate. I recommend revising this statement.

We agree and modified it as follows.

"Following injection of TME, rapid growth of *ind*-SOA was observed in each of the experiments".

3. The MAC$_{365}$ data as discussed seem to be primarily based on the findings from Figure 2, which were derived from methanol-extracted samples analyzed through Aquolog measurement. Notably, these values differ from those obtained via UPLC-PDA, as shown in Figure 4. It is essential for the author to offer a more comprehensive explanation for this discrepancy. Additionally, justification for the preference of Aquolog measurement data in this context would further enhance the clarity and validity of the findings.

The light absorption from Aqualog measurement was from methanol extracted samples However, the light absorption from UPLC-PDA measurement were from acetonitrile extracted samples. The different MAC values could be due to the different solvent. In addition, before UPLC-PDA measurement, the extracted samples were dried with pure nitrogen. This step could lead to vaporize volatile organic compounds. Furthermore, the Aqualog and UPLC-PDA are different instrumentational techniques. For Aqualog measurement, the attenuation was from the cuvette. For HUPLC-PDA measurement, the chromophores were separated by HPLC and then detected by a diode array detector. These could be the reasons why the MAC values are different between UPLC-PDA and Aqualog measurement. We added the following text in line 270-273 to explain this.

"Please note that the MAC values determined from Aqualog and UPLC-PDA measurements show differences (Figure 2 and 4). This could be caused by different solvent extraction, solution preparation, and instrumentational differences."

Reference

Lopez-Hilfiker, F. D., Mohr, C., Ehn, M., Rubach, F., Kleist, E., Wildt, J., Mentel, T. F., Lutz, A., Hallquist, M., Worsnop, D., and Thornton, J. A.: A novel method for online analysis of gas and particle composition: description and evaluation of a Filter Inlet for Gases and AEROsols (FIGAERO), Atmospheric Measurement Techniques, 7, 983-1001, 10.5194/amt-7-983-2014, 2014.